# MANIFOLD INSPIRED GRAPH CONTRASTIVE LEARNING

## ABSTRACT

Recently, graph contrastive learning (GCL) has emerged as a promising and trending paradigm for graph representation learning, providing generalizable node embeddings for various downstream tasks. However, current GCL methods often fail to fully exploit and encode the fine-grained graph structure information, leading to less informative node representations. In this study, we argue for a holistic approach that accounts for both node attributes and fine-grained graph structures, taking inspiration from spectral-based manifold learning techniques. Accordingly, we introduce MIGCL, a cutting-edge contrastive representation learning framework that employs cross-view adjacency reconstruction and feature orthogonalization. This dual approach not only retains the fine-grained graph/manifold structure information but also minimizes feature redundancy, thus averting the risk of representation collapse. To achieve feature orthogonalization, we employ an information-theoretic objective called Total Coding Rate. Our model can also be interpreted as a practical implementation of the Maximum Entropy Principle within the GCL context. Comprehensive experiments across three pivotal tasks: node classification, node clustering, and link prediction, affirm the method's efficacy and superiority. The implementation code can be found at: https://anonymous.4open.science/r/MIGCL-21C1.

## 1 INTRODUCTION

Self-supervised learning (SSL) has emerged as a promising paradigm for learning more informative representations without relying on human annotations. Typically, SSL models are pre-trained using carefully constructed pretext objectives, thereby serving as advantageous initializations for a wide range of downstream tasks (Liu et al., 2022b). In this sense, SSL has witnessed substantial advancements in the realm of graph representation learning, yielding performance, generalizability, and robustness metrics that are comparable, if not superior, to those achieved by supervised methods (Kipf & Welling, 2016; Veličković et al., 2019; Hassani & Khasahmadi, 2020; Chen et al., 2023).

A major branch of SSL methods is contrastive learning methods (Zhu et al., 2020; Zhang et al., 2021; Li et al., 2022a), which aim to learn representations by maximizing the agreement between two augmented instances. In the domains of computer vision (CV) and natural language processing (NLP), these methods involve two essential ingredients (Wang & Isola, 2020; Li et al., 2022b; Tong et al., 2023): (1) Representation of two different augmentations of the same instance should exhibit proximity (*invariance criterion*); (2) The latent representation must avoid collapsing into a trivial solution, e.g., all representations converge to a single point (*collapse avoidance*). While these criteria have been effective for CV and NLP tasks, their applicability to graph-structured data, which is inherently non-Euclidean, requires additional consideration. Given the topological interdependencies among nodes in graph-structured data, we introduce a third objective: models are supposed to capture essential information from the structural topology of graphs (*structure preservation*). This is essential for encapsulating the nuanced relational dynamics specific to graph-structured data.

Numerous recent studies (Veličković et al., 2019; Zhu et al., 2020; Zhang et al., 2021; Thakoor et al., 2022; Chen et al., 2023) have devised a variety of heuristics and strategies aimed at achieving the first two objectives, resulting in significant performance improvements. However, these studies achieve *structure preservation* either implicitly or at a rudimentary level. For example, methods such as GRACE (Zhu et al., 2020), CCA-SSG (Zhang et al., 2021), and BGRL (Thakoor et al., 2022)

tacitly integrate structure information by utilizing Graph Neural Networks as encoders. Meanwhile, DGI (Veličković et al., 2019) and MVGRL (Hassani & Khasahmadi, 2020) strive to encode global structure information by maximizing mutual information between (sub-)graph-level and node-level representations. Additional approaches like GCA (Zhu et al., 2021) and CSGCL (Chen et al., 2023) employ structure-aware augmentations to enhance representation learning. Moreover, works such as gCooL (Li et al., 2022a) and CSGCL (Chen et al., 2023) adopt a multi-view strategy contrasting node-level and community-level representations in order to capture higher-order community structures. Despite these efforts, none explicitly focus on capturing fine-grained structure information in graphs.

Graphs can be regarded as discrete analogs of continuous manifolds (Ni et al., 2019; Wu et al., 2023), wherein nodes correspond to manifold points and edges approximate geodesic paths between these points. Similar to graph representation learning, manifold learning aims to map the manifold-structured data points to a low-dimensional representation space while preserving the manifold structure (Li et al., 2022b).

Drawing inspiration from Laplacian Eigenmaps (LE), a classical spectral-based manifold learning method that preserves the geometric structure of the data pattern subject to the orthogonal condition of latent features (Belkin & Niyogi, 2003; Ghojogh et al., 2023), we introduce *Manifold Inspired Graph Contrastive Learning* (MIGCL). It follows the common practice of prior arts, generating two views of an input graph via random augmentation and obtaining node representations through a shared GNN encoder. Differently, our proposed approach harnesses a novel LE-motivated objective, where the cross-view adjacency reconstruction term preserves the fine-grained graph structure information, the feature orthogonalization term eliminates redundant correlations among features to prevent collapsed representation, and the view alignment term ensures that the learned representation remains invariant to augmentations of the same node instance. More specifically, we enforce the cross-view node similarity matrix to approximate graph adjacency matrix and utilize an information-theoretic objective Total Coding Rate (Ma et al., 2007) to achieve feature orthogonalization. Furthermore, our theoretical analysis sheds more lights that MIGCL can be seen as an instantiation of the Maximum Entropy Principle (Kesavan, 2009; Liu et al., 2022a) under the graph contrastive learning setting..

In summary, our primary contributions are:

- We propose three essential ingredients of graph contrastive learning and design a novel manifold learning inspired contrastive objective to satisfy these criteria.
- Formulation of a novel objective function incorporating cross-view adjacency reconstruction for fine-grained graph structure learning and the application of Total Coding Rate for feature orthogonalization. We further elucidate the theoretical foundations of our model via the Maximum Entropy Principle.
- We conduct a comprehensive evaluation of the proposed method on multiple real-world datasets and three downstream tasks, i.e., node classification, node clustering, and link prediction. The results demonstrate that our model achieves state-of-the-art performance.

## 2 BACKGROUND AND RELATED WORK

In this section, we first provide a formal description of the problem under investigation, followed by a literature review over manifold learning and graph contrastive learning.

### 2.1 PROBLEM STATEMENT

Let $\mathcal{G} = (\mathcal{V}, \mathcal{E})$ represent a graph, where $\mathcal{V} = \{v_1, v_2, \cdots, v_n\}$ and $\mathcal{E} \subseteq \mathcal{V} \times \mathcal{V}$ denote the node set and the edge set respectively. The graph $\mathcal{G}$ is associated with a feature matrix $\boldsymbol{X} \in \mathbb{R}^{n \times p}$, where $\boldsymbol{x}_i \in \mathbb{R}^p$ represents the feature of $v_i$, and an adjacency matrix $\boldsymbol{A} \in \{0, 1\}^{n \times n}$, where $\boldsymbol{A}_{i,j} = 1$ if and only if $(v_i, v_j) \in \mathcal{E}$. During training in the self-supervised setting, no task-specific labels are provided for $\mathcal{G}$. The goal is to learn an embedding function $f_\theta(\boldsymbol{A}, \boldsymbol{X})$ that maps $\boldsymbol{X}$ to $\boldsymbol{Z}$, where $\boldsymbol{Z} \in \mathbb{R}^{n \times d}$ and $d \ll p$. The pre-trained representations ought to encapsulate both *attribute* and *structure* information contained in $\mathcal{G}$ and can be easily transferable to various downstream tasks such as node classification, node clustering, and link prediction.

## 2.2 MANIFOLD LEARNING

Manifold models have sparked considerable interest for their geometric description of how data collected in high ambient dimension (e.g., pixels in an image) can vary with a few degrees of freedom. Following the hypothesis that high dimensional data tend to lie in the vicinity of a low dimensional manifold (Fefferman et al., 2013), manifold learning aims to map the manifold-structured data points to a low-dimensional representation space while preserving the manifold structure. Among various strategies, spectral-based manifold learning method, such as Laplacian Eigenmaps (LE), have garnered considerable attention. LE constructs a graph by linking proximal data points and then employs the eigenvectors of the graph's Laplacian matrix for data representation (Belkin & Niyogi, 2003; Ghojogh et al., 2023). Given that the adjacency matrix of a network inherently serves as the ground truth, LE is peculiarly well-suited for such contexts.

**Laplacian Eigenmaps.** Consider the dataset matrix $\boldsymbol{X} \in \mathbb{R}^{n \times p}$. We desire $d$-dimensional embeddings of data points $\boldsymbol{Z} \in \mathbb{R}^{n \times d}$ where $d \leq p$ and usually $d \ll p$. LE begins by constructing a proximity graph $\mathcal{G} = (\boldsymbol{A}, \boldsymbol{X})$, where the edge weights are built using one of many heuristics that determine which nodes are close to each other and can be binary or real-valued (Belkin & Niyogi, 2003). Examples of such heuristics include $k$-nearest neighbors, $\epsilon$-neighborhoods, heat kernels, etc. Subsequently, LE seeks to find a representation $\boldsymbol{Z}$ by minimizing the following optimization problem:

$$\min_{\boldsymbol{Z}} \sum_{i,j} \boldsymbol{A}_{i,j} ||\boldsymbol{z}_i - \boldsymbol{z}_j||^2 \text{ s.t. } \boldsymbol{Z}^\top \boldsymbol{Z} = \boldsymbol{I}. \tag{1}$$

Eq. 1 can be interpreted as follows: (1) If $\boldsymbol{x}_i$ and $\boldsymbol{x}_j$ are close to each other, the corresponding edge weight $\boldsymbol{A}_{i,j}$ is large. Consequently, minimizing the objective function involves minimizing the term $||\boldsymbol{z}_i - \boldsymbol{z}_j||^2$, which results in close $\boldsymbol{z}_i$ and $\boldsymbol{z}_j$. This aligns with the expectation that close data points should have similar embeddings. (2) Conversely, if $\boldsymbol{x}_i$ and $\boldsymbol{x}_j$ are far apart, $\boldsymbol{A}_{i,j}$ is small. Hence, the objective function is small due to multiplication by the small weight $\boldsymbol{A}_{i,j}$. Therefore, the embeddings $\boldsymbol{z}_i$ and $\boldsymbol{z}_j$ are not of significant concern, as the objective function is already minimized. (3) The constraint $\boldsymbol{Z}^\top \boldsymbol{Z} = \boldsymbol{I}$ is necessary to guarantee a non-trivial solution. Based on these three observations, LE effectively captures local data structure by prioritizing neighboring points while hoping to preserve the global structure through this localized fitting (Belkin & Niyogi, 2003). However, this local focus could be perceived as a limitation of LE (Ghojogh et al., 2023). In Section 3.2.3, we will discuss in more detail and alleviate this concern to learn fine-grained graph structure information.

## 2.3 GRAPH CONTRASTIVE LEARNING

In recent years, there has been significant advancement in the application of contrastive learning techniques for graph-structured data (Liu et al., 2022b). In this section, we provide a comprehensive overview of existing methods in graph contrastive learning (GCL). Our focus centers on the three essential ingredients delineated in Section 1.

**Invariance Criterion.** The first one is *invariance criterion*, designed to generate consistent embeddings for multiple views of the same instance. This criterion is generally implemented through similarity metrics such as cosine similarity or mean squared error between pairs of positive samples. From an optimization viewpoint, both approaches are functionally equivalent when employing normalized embeddings, thereby shifting our focus towards the remaining criteria that differentiate these methods.

**Collapse Avoidance.** The second one is *collapse avoidance*, an essential mechanism to inhibit the degeneracy of the latent space into trivial solutions. For instance, Veličković et al. (2019), Hassani & Khasahmadi (2020), Zhu et al. (2020), Zhu et al. (2021), Li et al. (2022a), and Chen et al. (2023) aim to distribute embeddings uniformly in the latent space by repelling negative sample pairs. BGRL (Thakoor et al., 2022), derived from BYOL (Grill et al., 2020), mitigates the collapse issue by employing asymmetric architectures inspired by distillation. CCA-SSG (Zhang et al., 2021), as well as our approach, revolve around regularization of the empirical covariance matrix of embeddings to prevent the occurrence of informational collapse in which the variables carry redundant information.

**Structure Preservation.** The last one is *structure preservation*, i.e., encoding graph structure information within the embeddings to the fullest extent possible. Nearly all existing methods employ

graph neural networks for implicit structural inclusion. Additionally, DGI (Veličković et al., 2019) and MVGRL (Hassani & Khasahmadi, 2020) focus on maximizing the mutual information between (sub-)graph-level representations and node-level representations to encode global structure information. GCA (Zhu et al., 2021) and CSGCL (Chen et al., 2023) utilize structure-aware augmentations to enrich representation learning. And, gCooL (Li et al., 2022a) and CSGCL (Chen et al., 2023) contrast node-level and community-level representations across two different views, aiming to capture the higher-order community structure of a graph. However, these methods tend to implicitly or coarsely capture graph structure information. Our work aims for a more explicit, fine-grained structure representation using a cross-view adjacency reconstruction objective.

## 3  Manifold Inspired Graph Contrastive Learning

### 3.1  Framework

Our model is elegantly constructed with three key components: 1) A random graph augmentation generator $\mathcal{T}$. 2) A GNN-based graph encoder symbolized as $f_\theta$, where $\theta$ representing its parameters. 3) A novel objective function, inspired by Laplacian Eigenmaps, to guide the optimization process.

**Graph Augmentation.**  The augmentation of graph data is a critical component of GCL, as it generates diverse graph views, leading to more generalized representations that are robust against variance (Liu et al., 2022b). In this study, we adopt the widely-used random graph augmentation pipeline, prevalent in previous works (Zhu et al., 2020; Zhang et al., 2021; Thakoor et al., 2022). Specifically, we employ feature masking and edge dropping to enhance both graph attributes and topological information. The function $\mathcal{T}$ comprises all possible graph transformation operations, and each $t \sim \mathcal{T}$ corresponds to a specific transformation applied to graph $\mathcal{G}$. It's worth noting that, to maintain focus on our novel objective function and ensure fair comparisons with existing methods, we use commonly-adopted augmentation techniques. However, it is possible to seamlessly integrate more complex random augmentations (Zhu et al., 2021; Chen et al., 2023) into our framework. Further details regarding the employed augmentation functions can be found in Appendix B.2.

**Training.**  During each training epoch, we first sample two random augmentation functions $t^1 \sim \mathcal{T}$ and $t^2 \sim \mathcal{T}$, and then generate two views $\widetilde{\mathcal{G}}^1 = (\widetilde{A}^1, \widetilde{X}^1)$ and $\widetilde{\mathcal{G}}^2 = (\widetilde{A}^2, \widetilde{X}^2)$ based on the chosen functions. These two views are subsequently fed into the shared encoder $f_\theta$, including a 2-layer GCN (Kipf & Welling, 2017) and a 2-layer MLP, to extract the corresponding node embeddings: $Z^1 = f_\theta(\widetilde{A}^1, \widetilde{X}^1)$ and $Z^2 = f_\theta(\widetilde{A}^2, \widetilde{X}^2)$. Similar to Wang & Isola (2020), we further $\ell_2$-normalize the node embeddings to reside on the unit hypersphere, which is also a prerequisite for feature orthogonalization in our model (refer to Appendix C.2). To help better understand the proposed method, we provide the PyTorch-style pseudocode for training MIGCL in Algorithm 1.

**Inference.**  To obtain node embeddings for downstream tasks, the original graph $\mathcal{G} = (A, X)$ is fed into the trained encoder $f_\theta$, yielding $Z = f_\theta(A, X)$. Before applying them to downstream tasks, these embeddings are also $\ell_2$-normalized.

### 3.2  Manifold Inspired Learning Objective

In this section, we elaborate on our manifold-inspired contrastive objective and demonstrate how it satisfies the three essential ingredients of GCL outlined in Section 1.

#### 3.2.1  Total Coding Rating as Feature Orthogonalization

Corresponding to the feature orthogonalization restriction in Laplacian Eigenmaps (see Section 2.2), we adopt an information-theoretic objective Total Coding Rating (TCR) (Ma et al., 2007; Liu et al., 2022a; Han et al., 2022; Li et al., 2022b), to orthogonalize features and prevent collapsed representations. TCR approximates the coding length in lossy data compression, and can be seen as a computationally feasible alternative for quantifying the entropy of continuous random variables. Intuitively, maximizing TCR aligns with our objective of increasing the informativeness of learned representations. Formally, given the representations $Z \in \mathbb{R}^{n \times d}$ and an allowable distortion $\sqrt{\epsilon}$,

TCR is defined to be:

$$\text{TCR}(\boldsymbol{Z}) = \frac{1}{2} \log \det \left( \boldsymbol{I} + \frac{d}{n\epsilon} \boldsymbol{Z}\boldsymbol{Z}^\top \right). \tag{2}$$

Based on straightforward reasoning (Appendix C.2), it becomes evident that maximizing TCR promotes the orthogonalization of latent features. Accordingly, we calculate the feature orthogonalization loss for the generated graph views.

$$\mathcal{L}_{ort} = -\frac{1}{2} \left( \text{TCR}(\boldsymbol{Z}^1) + \text{TCR}(\boldsymbol{Z}^2) \right). \tag{3}$$

### 3.2.2 Combining with the View Invariance Prior

In contrastive learning, the *invariance criterion* requires that different augmentations of the same instance be close in the representation space (Wang & Isola, 2020). We maximize the cosine similarity between multi-view node representations. With $\ell_2$-normalized $\boldsymbol{Z}^1$ and $\boldsymbol{Z}^2$, the invariance loss is:

$$\mathcal{L}_{inv} = -\frac{1}{n} \sum_i (\boldsymbol{z}_i^1 \cdot \boldsymbol{z}_i^2), \tag{4}$$

where $\boldsymbol{z}_i^1$ and $\boldsymbol{z}_i^2$ are the representations of node $v_i$ in different views, and $\cdot$ represents the dot product operation.

### 3.2.3 Cross-view Adjacency Matrix Reconstruction

Laplacian Eigenmaps aims to embed nearby points closely, emphasizing local relationships (Section 2.2) (Ghojogh et al., 2023). This focus can obscure global structures, affecting downstream tasks negatively (Table 4). To address this, our cross-view strategy, analogous to graph auto-encoders (Kipf & Welling, 2016), balances local and global relationships. We define the structure preservation loss as follows:

$$\mathcal{L}_{str} = -\frac{1}{n^2 - n} \sum_i \left( \sum_{j \in \mathcal{N}_i} \log \sigma(\boldsymbol{z}_i^1 \cdot \boldsymbol{z}_j^2) + \sum_{j \notin \mathcal{N}_i} \log \left( 1 - \sigma(\boldsymbol{z}_i^1 \cdot \boldsymbol{z}_j^2) \right) \right), \tag{5}$$

Here, $\mathcal{N}_i$ represents the neighbors of node $v_i$, and $\sigma(\cdot)$ denotes the sigmoid function: $\sigma(x) = 1/(1 + e^{-x})$. This objective cross-viewly pulls together connected nodes and pushes away disconnected nodes. Though distancing all non-neighboring nodes may seem counterintuitive, our ablation study substantiates its importance (Table 4). We acknowledge that controlling the selection order of neighbors could be another dimension to explore, but it falls outside the scope of this paper.

### 3.2.4 Overall Objective

Combining collapse avoidance, invariance criterion and structure preservation, we formulate our overall objective as follows:

$$\mathcal{L} = \mathcal{L}_{ort} + \lambda \mathcal{L}_{inv} + \gamma \mathcal{L}_{str}, \tag{6}$$

where $\lambda$ and $\gamma$ are weights that balance the contributions of three terms $\mathcal{L}_{ort}$, $\mathcal{L}_{inv}$ and $\mathcal{L}_{str}$.

**Computational Complexity.** The computation of the $\mathcal{L}_{ort}$ term is relatively efficient due to the commutative property of TCR: $\text{TCR}(\boldsymbol{Z}) = \frac{1}{2} \log \det \left( \boldsymbol{I} + \frac{d}{n\epsilon} \boldsymbol{Z}\boldsymbol{Z}^\top \right) = \frac{1}{2} \log \det \left( \boldsymbol{I} + \frac{d}{n\epsilon} \boldsymbol{Z}^\top \boldsymbol{Z} \right)$. Given $\boldsymbol{Z} \in \mathbb{R}^{n \times d}$, the matrices $\boldsymbol{Z}\boldsymbol{Z}^\top \in \mathbb{R}^{n \times n}$ and $\boldsymbol{Z}^\top \boldsymbol{Z} \in \mathbb{R}^{d \times d}$ have different dimensions. Although the computation of $\log \det \left( \boldsymbol{I} + \frac{d}{n\epsilon} \boldsymbol{Z}\boldsymbol{Z}^\top \right)$ incurs a time complexity of $\mathcal{O}(n^3)$, using $\log \det \left( \boldsymbol{I} + \frac{d}{n\epsilon} \boldsymbol{Z}^\top \boldsymbol{Z} \right)$ reduces the complexity to $\mathcal{O}(d^3)$, and typically $d \ll n$. In our experiments, we consider $d$ from $64, 128, 256$, making the $\log \det(\cdot)$ operation run in constant time with respect to $n$. The complexities for computing $\mathcal{L}_{inv}$ and $\mathcal{L}_{str}$ are $\mathcal{O}(n)$ and $\mathcal{O}(n^2 - n)$, respectively.

### 3.2.5 Theoretical Insights with Connection to Information Theory

*What makes for generalizable and informative representations?* This question can be tackled through the maximum entropy principle in information theory (Kesavan, 2009; Liu et al., 2022a). This principle posits that the most informative probability distribution is the one with highest entropy, devoid of added bias given a set of testable information. Translating this to graph contrastive

learning, our hypothesis posits that an effective representation should aim to maximize entropy within the bounds set by data augmentation invariance and graph structure. Here, TCR serves as a computable proxy for entropy. Consequently, our model operationalizes the maximum entropy principle by striving to maximize the entropy of the representations through $\mathcal{L}_{ort}$, while still minimizing $\mathcal{L}_{inv}$ and $\mathcal{L}_{str}$ to respect the given priors.

# 4 EXPERIMENTS

## 4.1 EXPERIMENTAL SETTINGS

**Datasets.** We assess the efficacy of our pre-trained embeddings across three key tasks: node classification, node clustering, and link prediction. The evaluations are conducted on four diverse datasets—Cora, WikiCS, Photo, and Computer. Dataset specifics are elaborated in Appendix B.1.

**Baselines.** We benchmark MIGCL against seven contemporary graph contrastive learning algorithms: DGI (Veličković et al., 2019), MVGRL (Hassani & Khasahmadi, 2020), GRACE (Zhu et al., 2020), GCA (Zhu et al., 2021), CCA-SSG (Zhang et al., 2021), gCooL (Li et al., 2022a), and CSGCL (Chen et al., 2023). Additionally, we include Laplacian Eigenmaps (Belkin & Niyogi, 2003), VGAE Kipf & Welling (2016) and raw feature baselines in our evaluation.

**Evaluation Protocol.** Following the evaluation protocol in (Zhu et al., 2020), we first train unsupervised models, then freeze encoder parameters for feature extraction. For node classification, we employ logistic regression on twenty 1:1:8 train/validation/test random node splits for Cora, Photo, and Computer, and use public splits for WikiCS. Node clustering is performed using KMeans, executed twenty times. For link prediction, dot products between node pair representations are fed into a logistic regression decoder, trained on twenty 8.5:0.5:1 train/validation/test random edge splits, as described in Kipf & Welling (2016).

**Metrics.** Evaluation metrics include Micro-F1 and Macro-F1 for node classification; NMI and ARI for node clustering; and AUC and AP for link prediction. Each metric is presented with its standard deviation for robust assessment.

**Implementation Details.** We employ the official GitHub implementations for baseline methods and build MIGCL in PyTorch. The encoder $f_\theta$ uses a 2-layer GCN (Kipf & Welling, 2017) followed by a 2-layer MLP. Distortion measure $\epsilon$ in Eq. 2 is 0.01. ELU activation (Clevert et al., 2015) and Adam optimizer (Kingma & Ba, 2014) are used across datasets. For dataset-specific hyperparameters, see Appendix B.3.

## 4.2 EXPERIMENTAL RESULTS

**Evaluation on Node Classification and Node Clustering.** The empirical results on node classification and node clustering are presented in Table 1 and Table 2, respectively. These results demonstrate that our proposed approach outperforms state-of-the-art methods across all four datasets. For example, on the Photo dataset, our approach improves NMI and ARI by approximately 5% compared to the previous state-of-the-art method, GRACE. This superiority can be attributed to three key factors: maximally diverse representation, intra-class/cluster compactness, and inter-class/cluster separation. Firstly, the optimization of TCR encourages the expansion of the latent space, facilitating the acquisition of highly diverse representations. Based on the homophily assumption, the invariance criterion and the pulling together of connected node pairs contribute to increased compactness within classes/clusters. Moreover, the pushing away of disconnected node pairs promotes inter-class/cluster separation. These three properties of the learned representation significantly benefit node classification and node clustering tasks.

**Evaluation on Link Prediction.** Table 3 presents our link prediction performance, which holds its own against competing methods. This outcome is largely due to our novel cross-view adjacency matrix reconstruction objective. By prioritizing the preservation of fine-grained structure information in the graph, our approach is well-suited to meet the specific demands of link prediction tasks.

**Ablation Study.** To systematically examine the influence of each loss component, we conduct ablation studies with varying combinations of $\mathcal{L}_{ort}$ (Eq. 3), $\mathcal{L}_{inv}$ (Eq. 4), $\mathcal{L}_{str}$ (Eq. 5), and $\mathcal{L}_{str}$'s

Table 1: Overall performance on node classification (in percentage).

| Dataset | Cora | | WikiCS | | Photo | | Computer | |
|---------|------|------|--------|------|-------|------|----------|------|
| Metric | MiF1 | MaF1 | MiF1 | MaF1 | MiF1 | MaF1 | MiF1 | MaF1 |
| RawFeat | 55.8±2.5 | 49.0±3.6 | 72.9±0.5 | 69.3±0.8 | 83.1±1.6 | 78.6±2.9 | 77.6±1.0 | 67.4±1.9 |
| LE | 76.2±2.3 | 74.3±2.9 | 73.8±0.8 | 70.9±0.9 | 87.7±1.1 | 85.7±1.3 | 84.1±1.0 | 83.7±1.3 |
| VGAE | 78.4±2.4 | 77.9±0.5 | 75.1±0.7 | 67.0±1.0 | 91.7±1.1 | 90.3±1.4 | 87.6±0.9 | 86.8±1.0 |
| DGI | 83.2±2.1 | 82.0±2.4 | 78.3±0.6 | 75.2±0.7 | 91.7±1.1 | 90.4±1.3 | 88.0±0.8 | 86.7±0.9 |
| MVGRL | 83.5±2.5 | 81.5±2.7 | 77.6±0.5 | 74.3±0.7 | 92.0±1.0 | 90.5±1.4 | 87.4±0.9 | 85.5±1.3 |
| GRACE | 83.2±2.3 | 81.4±2.7 | 78.6±0.3 | 75.7±0.5 | 92.4±1.0 | 91.2±1.3 | 88.8±0.9 | 87.1±1.1 |
| GCA | 82.6±0.7 | 81.2±0.7 | 79.1±0.7 | 76.3±1.1 | 92.2±1.1 | 90.8±1.5 | 88.8±0.9 | 87.8±1.0 |
| CCA-SSG | 83.5±2.8 | 82.2±3.1 | 78.0±0.7 | 74.8±0.8 | 92.9±1.0 | 91.6±1.2 | 89.2±0.6 | 88.1±1.1 |
| gCooL | 82.8±2.0 | 81.7±2.3 | 79.0±0.4 | 76.3±0.6 | 92.4±1.1 | 90.9±1.7 | 88.5±0.6 | 87.3±1.4 |
| CSGCL | 83.3±2.1 | 82.1±2.5 | 79.0±0.7 | 76.1±0.9 | 92.7±0.9 | 91.5±1.0 | 89.7±0.7 | 88.6±1.2 |
| MIGCL | **84.5±2.3** | **83.0±2.5** | **79.5±0.3** | **77.1±0.5** | **93.2±0.9** | **92.0±1.2** | **89.9±1.1** | **88.8±1.1** |

Table 2: Overall performance on node clustering (in percentage).

| Dataset | Cora | | WikiCS | | Photo | | Computer | |
|---------|------|------|--------|------|-------|------|----------|------|
| Metric | NMI | ARI | NMI | ARI | NMI | ARI | NMI | ARI |
| RawFeat | 15.3±3.4 | 9.5±1.9 | 26.4±0.3 | 15.5±0.4 | 32.6±0.4 | 20.7±0.9 | 24.3±0.5 | 9.4±0.4 |
| LE | 45.8±0.7 | 34.0±1.2 | 38.7±1.6 | 28.7±2.0 | 51.2±2.0 | 33.7±2.5 | 47.0±1.3 | 32.4±2.7 |
| VGAE | 52.8±1.1 | 44.5±1.5 | 43.2±0.3 | 32.4±1.1 | 65.7±0.0 | 56.9±0.0 | 50.2±1.2 | 30.3±1.8 |
| DGI | 52.3±0.2 | 43.1±0.5 | 40.9±0.4 | 29.0±1.4 | 64.6±0.0 | 55.3±0.0 | 51.7±1.1 | 39.5±0.8 |
| MVGRL | 51.0±0.4 | 46.0±0.7 | 38.3±0.6 | 26.8±1.1 | 65.7±1.0 | 55.6±2.7 | 50.1±0.5 | 37.3±1.9 |
| GRACE | 52.8±1.1 | 46.3±1.8 | 47.2±0.5 | 40.3±0.6 | 67.6±0.8 | 57.1±0.9 | 51.9±0.1 | 32.6±0.9 |
| GCA | 49.0±0.4 | 41.1±0.2 | 46.5±0.9 | 39.0±1.3 | 64.5±0.4 | 55.3±1.5 | 53.0±1.3 | 32.0±1.9 |
| CCA-SSG | 55.1±1.6 | 47.0±2.2 | 45.4±0.1 | 36.7±0.0 | 63.8±0.0 | 54.7±0.0 | 52.1±0.1 | 37.6±0.2 |
| gCooL | 50.2±1.2 | 44.7±2.4 | 47.1±0.0 | 39.0±0.1 | 64.5±0.0 | 55.5±0.0 | 49.6±0.2 | 28.4±0.7 |
| CSGCL | 53.1±1.5 | 48.2±1.5 | 46.5±1.2 | 37.6±3.2 | 61.0±0.0 | 52.4±0.0 | 55.3±0.8 | 34.2±1.4 |
| MIGCL | **57.3±0.4** | **51.2±1.8** | **47.5±0.5** | **41.2±0.8** | **72.5±0.0** | **62.3±0.0** | **58.9±1.3** | **40.8±1.7** |

variant $\mathcal{L}'_{str}$ [1]. We assess their impact on node classification, node clustering, and link prediction across the Cora and Photo datasets. The outcomes are collated in Table 4 with the following key observations: (1) Introducing the feature orthogonalization term $\mathcal{L}_{ort}$ notably improves node classification metrics by facilitating more discriminative representations and mitigating feature collapse. (2) Implementing the structure preservation term $\mathcal{L}_{str}$ significantly boosts node clustering results, attesting to its capacity to encode fine-grained graph structure information into node representations effectively. (3) Opting for a standalone strategy of merely pulling nodes together, exemplified by $\mathcal{L}'_{str}$, results in a discernable performance dip, supporting the arguments laid out in Section 3.2.3. In summary, the composite inclusion of all terms in our objective function is crucial for optimizing performance. This integration not only enables MIGCL to acquire representations rich in fine-grained graph structure information but also reduces feature redundancy. Driven by the three ingredients of graph contrastive learning, i.e., invariance criterion, collapse avoidance, and structure preservation, our model succeeds in generating highly generalizable and informative node representations.

---

[1] $\mathcal{L}'_{str}$ only cross-viewly pulls together connected nodes, i.e., $\mathcal{L}'_{str} = -\frac{1}{|\mathcal{E}|} \sum_i \sum_{j \in \mathcal{N}_i} \log \sigma(z_i^1 \cdot z_j^2)$, while $\mathcal{L}_{str}$ also pushes away disconnected nodes.

Table 3: Overall performance on link prediction (in percentage).

| Dataset | Cora | | WikiCS | | Photo | | Computer | |
|---|---|---|---|---|---|---|---|---|
| Metric | AUC | AP | AUC | AP | AUC | AP | AUC | AP |
| RawFeat | 83.1±0.8 | 85.1±0.8 | 92.9±0.1 | 92.6±0.1 | 86.1±0.2 | 84.6±0.3 | 85.3±0.1 | 83.7±0.2 |
| LE | 84.4±0.8 | 88.1±0.8 | 94.2±0.1 | 94.6±0.1 | 96.9±0.1 | 96.2±0.1 | 94.5±0.1 | 93.3±0.2 |
| VGAE | 92.5±0.6 | 93.3±0.6 | 97.6±0.0 | 97.5±0.0 | 97.1±0.1 | 97.0±0.1 | 97.3±0.0 | 97.3±0.1 |
| DGI | 93.9±0.5 | 94.5±0.6 | 98.0±0.1 | 97.9±0.1 | 97.6±0.1 | 97.3±0.1 | 97.2±0.1 | 96.9±0.1 |
| MVGRL | 92.9±0.6 | 93.4±0.7 | 97.5±0.1 | 97.4±0.1 | 97.1±0.1 | 96.9±0.1 | 96.5±0.1 | 96.3±0.1 |
| GRACE | 95.1±0.5 | 95.2±0.5 | 98.0±0.0 | 97.8±0.1 | **97.9±0.1** | 97.3±0.2 | 96.9±0.0 | 96.1±0.1 |
| GCA | 94.4±0.8 | 94.7±0.9 | 97.2±0.1 | 97.2±0.1 | 95.4±0.1 | 94.3±0.2 | 97.3±0.0 | 96.6±0.1 |
| CCA-SSG | 95.1±0.5 | 95.4±0.6 | 98.0±0.1 | 97.9±0.1 | 97.3±0.1 | 96.7±0.2 | 96.9±0.0 | 96.6±0.1 |
| gCooL | 94.7±0.5 | 95.2±0.5 | 97.9±0.1 | 97.8±0.1 | 95.8±0.1 | 94.9±0.2 | 97.4±0.1 | 96.7±0.1 |
| CSGCL | **95.3±0.5** | **95.8±0.6** | 92.9±0.1 | 94.7±0.1 | 94.7±0.1 | 93.7±0.2 | 97.8±0.0 | 97.2±0.1 |
| MIGCL | 94.9±0.4 | 95.5±0.5 | **98.1±0.0** | **98.1±0.1** | 97.7±0.1 | **97.5±0.1** | **97.9±0.0** | **97.7±0.1** |

Table 4: Ablation study on three downstream tasks, comparing combinations of objectives.

| Variants | Cora | | | Photo | | |
|---|---|---|---|---|---|---|
| | MiF1 | NMI | AUC | MiF1 | NMI | AUC |
| $\mathcal{L}_{ort} + \mathcal{L}_{inv}$ | 81.89±2.14 | 42.13±4.11 | 90.86±0.60 | 92.76±0.79 | 62.29±3.84 | 97.35±0.17 |
| $\mathcal{L}_{str} + \mathcal{L}_{inv}$ | 79.91±3.06 | 50.02±0.19 | 91.71±0.70 | 91.90±1.06 | 67.43±0.81 | 95.61±0.12 |
| $\mathcal{L}_{ort} + \mathcal{L}_{inv} + \mathcal{L}'_{str}$ | 82.35±1.82 | 41.50±7.04 | 89.66±0.63 | 92.36±0.73 | 55.92±1.65 | 97.41±0.14 |
| $\mathcal{L}_{ort} + \mathcal{L}_{inv} + \mathcal{L}_{str}$ | 84.47±2.27 | 57.30±0.40 | 94.90±0.43 | 93.21±0.85 | 72.53±0.01 | 97.66±0.09 |

**Visualization of Node Representations.** For a more intuitive understanding of the node representations, we employ t-SNE (van der Maaten & Hinton, 2008) for dimensionality reduction. As delineated in Figure 1, the MIGCL-generated representations exhibit marked intra-cluster compactness and inter-cluster separation, thus underscoring the efficacy of our approach.

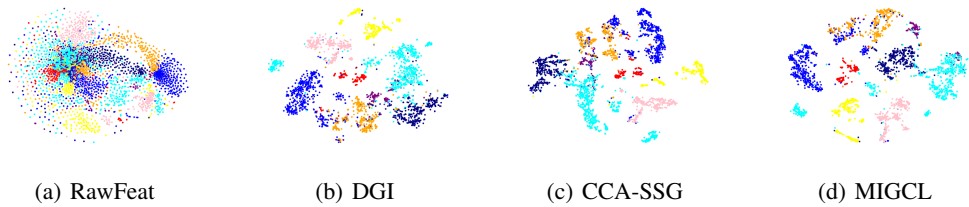

(a) RawFeat          (b) DGI          (c) CCA-SSG          (d) MIGCL

Figure 1: t-SNE visualization of representations on Photo.

**Effects of Weights $\lambda$ and $\gamma$.** We investigate the impact of the intensities of the invariance criterion term and structure preservation term on performance by varying the trade-off hyperparameters $\lambda$ and $\gamma$. Performance outcomes for node classification, clustering, and link prediction are illustrated in Figures 2 and 3 for Cora and Photo datasets, respectively. From these figures, we observe that decreasing $\lambda$ usually leads to a decrease in performance. Moreover, the lower-right half of the heatmap generally outperforms the other half. The intuition behind these observations is that while not all connected nodes truly belong to the same class, we can reasonably assume that different views of the same node do belong to the same class. As a result, we recommend setting $\lambda$ higher than $\gamma$ to prioritize the invariance criterion loss.

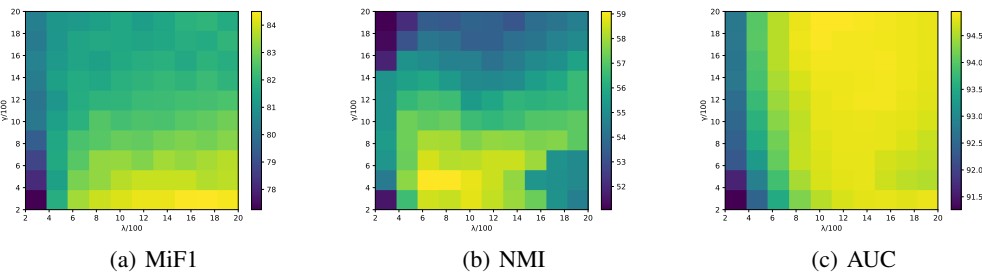

(a) MiF1          (b) NMI          (c) AUC

Figure 2: Visualization of the effects of $\lambda$ and $\gamma$ on Cora.

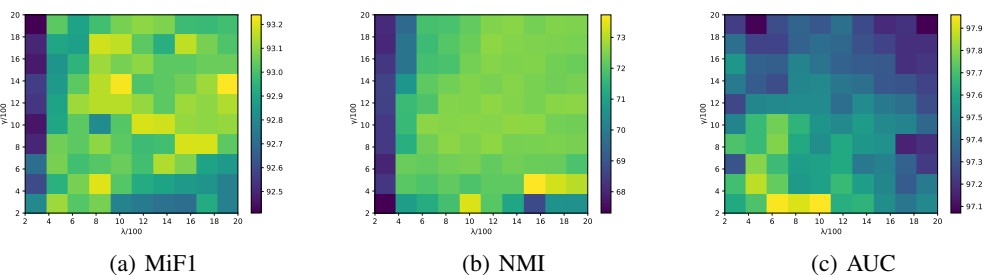

(a) MiF1          (b) NMI          (c) AUC

Figure 3: Visualization of the effects of $\lambda$ and $\gamma$ on Photo.

## 5 CONCLUSION

In this study, we identify three essential ingredients of graph contrastive learning: *invariance criterion*, *collapse avoidance*, and *structure preservation*. These tenets inform the design of our novel objective function, inspired by manifold learning. Leveraging total coding rate, we achieve feature orthogonalization, while our cross-view adjacency matrix reconstruction captures fine-grained graph structures. Moreover, we offer an information-theoretic lens to understand our model through the Maximum Entropy Principle. Comprehensive experimental evaluations across node classification, node clustering, and link prediction affirm the model's superior performance.

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

## A    ALGORITHM

---

**Algorithm 1** PyTorch-style pseudocode for MIGCL

---

```
# A: adjacency matrix
# X: node features
# f: encoder network
# lambda, gamma: trade-off
#
# tcr: function to calculate total coding rate
# off_diagonal: off-diagonal elements of a matrix

# generate two views through random augmentation
A1, X1 = augment(A, X)
A2, X2 = augment(A, X)

# compute l2-normalized embeddings
Z1 = normalize(f(A1, X1))
Z2 = normalize(f(A1, X1))

# compute feature orthogonalization loss
loss_ort = - (tcr(Z1) + tcr(Z2)) / 2

# compute cross-view node cosine similarity matrix
Gram = mm(Z1, Z2.T)

# compute invariance criterion loss
loss_inv = - trace(Gram) / X.shape[0]

# compute structure preservation loss
Gram_ = sigmoid(off_diagonal(Gram))
A_ = off_diagonal(A)
loss_str = binary_cross_entropy(Gram_, A_)

loss = loss_ort + lambda * loss_inv + gamma * loss_str
```

---

## B    IMPLEMENTATION DETAILS

Table 5: Dataset statistics.

| Dataset | Type | #Nodes | #Edges | #Features | #Classes |
|---|---|---|---|---|---|
| Cora | citation | 2,708 | 5,429 | 1,433 | 7 |
| WikiCS | reference | 11,701 | 216,123 | 300 | 10 |
| Photo | co-purchase | 7,650 | 119,081 | 745 | 8 |
| Computer | co-purchase | 13,752 | 245,861 | 767 | 10 |

### B.1    DATASETS

We evaluate our model on four representative datasets: Cora, WikiCS, Photo and Computer. Their detailed statistics are summarized in Table 5, and brief introductions are as follows:

- **Cora** (Kipf & Welling, 2017) is a widely recognized citation network dataset where nodes correspond to publications, and edges depict the citations between them. Each node is labeled based on the subject matter of the respective paper.

- **WikiCS** (Mernyei & Cangea, 2020) is a reference network constructed from Wikipedia. It comprises nodes that correspond to articles in the field of Computer Science, where edges are derived from hyperlinks. The dataset encompasses 10 distinct classes that represent various branches within the field. Node features are computed as the average GloVe word embeddings of the respective articles.

- **Photo** and **Computer** (Shchur et al., 2018) are two networks derived from Amazon's co-purchase relationships. In these networks, nodes represent goods, edges indicate frequent co-purchases between goods, node features are represented by bag-of-words encoded product reviews, and class labels are assigned based on the respective product categories.

For all datasets, we use the processed version provided by Deep Graph Library (Wang et al., 2019). All datasets are public available and do not have licenses.

## B.2 Graph Augmentation

We employ two graph data augmentation strategies designed to enhance graph attributes and topology information, respectively. They are widely used in node-level contrastive learning (Zhu et al., 2020; Zhang et al., 2021; Thakoor et al., 2022).

**Feature Masking (FM).** We randomly select a portion of the node features' dimensions and mask their elements with zeros. Formally, we first sample a random vector $\widetilde{m} \in \{0,1\}^F$, where each dimension is drawn from a Bernoulli distribution with probability $1 - p_m$, i.e., $\widetilde{m}_i \sim \mathcal{B}(1 - p_m), \forall i$. Then, the masked node features $\widetilde{X}$ are computed by $\|_{i=1}^{N} x_i \odot \widetilde{m}$, where $\odot$ denotes the Hadamard product and $\|$ represents the stack operation (i.e., concatenating a sequence of vectors along a new dimension).

**Edge Dropping (EP).** In addition to feature masking, we stochastically drop a certain fraction of edges from the original graph. Formally, since we only remove existing edges, we first sample a random masking matrix $\widetilde{D} \in \{0,1\}^{N \times N}$, with entries drawn from a Bernoulli distribution $\widetilde{D}_{i,j} \sim \mathcal{B}(1 - p_d)$ if $A_{i,j} = 1$ for the original graph, and $\widetilde{D}_{i,j} = 0$ otherwise. Here, $p_d$ represents the probability of each edge being dropped. The corrupted adjacency matrix can then be computed as $\widetilde{A} = A \odot \widetilde{D}$.

Similar to previous works (Zhu et al., 2020; 2021; Thakoor et al., 2022; Li et al., 2022a; Chen et al., 2023), we jointly utilize these two methods to generate graph views. To provide different contexts in the two views, the generation is controlled by two distinct sets of hyperparameters: $p_{m1}, p_{d1}$ and $p_{m2}, p_{d2}$.

Table 6: Hypeparameter specifications of node classification and clustering.

| Dataset | $p_{d1}, p_{d2}, p_{m1}, p_{m2}$ | $\lambda, \gamma$ | lr, wd | #hid_units | #epochs |
|---|---|---|---|---|---|
| Cora | 0.2, 0.4, 0.0, 0.2 | 1600, 200 | 1e-4, 1e-4 | 256-128 | 400 |
| WikiCS | 0.0, 0.6, 0.1, 0.0 | 1000, 50 | 1e-3, 1e-5 | 256-128 | 450 |
| Photo | 0.8, 0.8, 0.0, 0.0 | 1000, 1400 | 1e-3, 1e-5 | 128-64 | 450 |
| Computer | 0.6, 0.8, 0.0, 0.1 | 800, 600 | 1e-3, 1e-5 | 256-256 | 1500 |

## B.3 Detailed Hyperparameters

Detailed hyperparameters for both node classification and node clustering can be found in Table 6. Notably, the hyperparameters for link prediction differ due to the corruption of the original graph structure (see Section 4.1), and these specific hyperparameters are listed in Table 7.

Table 7: Hypeparameter specifications of link prediction.

| Dataset | $p_{d1}, p_{d2}, p_{m1}, p_{m2}$ | $\lambda, \gamma$ | lr, wd | #hid_units | #epochs |
|---|---|---|---|---|---|
| Cora | 0.2, 0.1, 0.0, 0.2 | 1000, 1800 | 1e-3, 1e-5 | 256-128 | 400 |
| WikiCS | 0.0, 0.6, 0.1, 0.0 | 1500, 50 | 1e-3, 1e-5 | 256-128 | 400 |
| Photo | 0.5, 0.4, 0.0, 0.0 | 1000, 200 | 1e-3, 1e-5 | 128-64 | 500 |
| Computer | 0.6, 0.8, 0.0, 0.1 | 800, 10 | 1e-3, 1e-5 | 256-256 | 1500 |

# C  UNDERSTANDING THE TOTAL CODING RATE

## C.1  INSIGHTS OF THE TOTAL CODING RATE

We first present how to derive the total coding rate of entire node representations following Ma et al. (2007). Suppose we have an embedding matrix $\boldsymbol{Z} \in \mathbb{R}^{n \times d}$, and let $\epsilon$ be the error allowable for encoding every vector $\boldsymbol{z}_i$ in $\boldsymbol{Z}$. In other words, we are allowed to distort each vector $\boldsymbol{z}_i$ with random variable $\boldsymbol{v}_i$ of variance $\frac{\epsilon}{d}$. So we have

$$\hat{\boldsymbol{z}}_i = \boldsymbol{z}_i + \boldsymbol{v}_i, \text{with } \boldsymbol{v}_i = \mathcal{N}(0, \frac{\epsilon}{d}\boldsymbol{I}), \tag{7}$$

Then the covariance matrix of $\boldsymbol{z}_i$ is

$$\mathbb{E}\left[\frac{1}{n}\sum_{i=1}^{n}\hat{\boldsymbol{z}}_i\hat{\boldsymbol{z}}_i^\top\right] = \frac{\epsilon}{d}\boldsymbol{I} + \frac{1}{n}\boldsymbol{Z}\boldsymbol{Z}^\top, \tag{8}$$

And the volumes of covariance matrix and random vector $\boldsymbol{v}_i$ are

$$\text{vol}(\hat{\boldsymbol{Z}}) \propto \sqrt{\det\left(\frac{\epsilon}{d}\boldsymbol{I} + \frac{1}{n}\boldsymbol{Z}\boldsymbol{Z}^\top\right)},$$
$$\text{vol}(\boldsymbol{v}) \propto \sqrt{\det\left(\frac{\epsilon}{d}\boldsymbol{I}\right)}, \tag{9}$$

Then the number of bit needed to encode the $\boldsymbol{Z}$ is

$$\text{TCR}(\boldsymbol{Z}) = \log_2\left(\frac{\text{vol}(\hat{\boldsymbol{Z}})}{\text{vol}(\boldsymbol{v})}\right) = \frac{1}{2}\log_2\det\left(\boldsymbol{I} + \frac{d}{n\epsilon}\boldsymbol{Z}\boldsymbol{Z}^\top\right). \tag{10}$$

## C.2  TOTAL CODING RATING AS FEATURE ORTHOGONALIZATION

Here we demonstrate that the optimization of Eq. 3 is equivalent to feature orthogonalization.

*Proof.* We start by considering the property of the $\log\det$ function (Kang et al., 2015) and the property of the Frobenius norm. For any matrix $\boldsymbol{Z} \in \mathbb{R}^{n \times d}$, where $d \leq n$, the following hold:

$$\log\det\left(\boldsymbol{I} + a\boldsymbol{Z}\boldsymbol{Z}^\top\right) = \sum_{i=1}^{d}\log\left(1 + a\sigma_i^2\right), \tag{11}$$

$$||\boldsymbol{Z}||_{\text{F}}^2 = \sum_{i=1}^{d}\sigma_i^2, \tag{12}$$

where $a$ is a real value, and $\sigma_i$ represents the $i$-th singular value of the matrix $\boldsymbol{Z}$.

Next, let us consider the following optimization problem:

$$\max\sum_{i=1}^{d}\log(1 + x_i) \text{ s.t. } \sum_{i=1}^{d}x_i = C. \tag{13}$$

As the function $\log(1 + x)$ is concave, the optimization of Eq. 13 reaches its maximum when each $x_i$ is equal to $C/d$.

Since the rows of $\boldsymbol{Z}$ are $\ell_2$-normalized, i.e., $||\boldsymbol{Z}||_{\text{F}}^2 = n$, we can reformulate the optimization problem in Eq. 2 as:

$$\max\sum_{i=1}^{d}\log\left(1 + \frac{d}{n\epsilon}\sigma_i^2\right) \text{ s.t. } \sum_{i=1}^{d}\sigma_i^2 = n. \tag{14}$$

Optimizing Eq. 14 yields a solution where each $\sigma_i^2$ is equal to $\frac{n}{d}$. This uniform distribution of singular values corresponds to diagonal covariance, i.e., $\boldsymbol{Z}^\top \boldsymbol{Z} = \frac{n}{d} \boldsymbol{I}$. As we can obtain $\left( \sqrt{\frac{d}{n}} \boldsymbol{Z} \right)^\top \left( \sqrt{\frac{d}{n}} \boldsymbol{Z} \right) = \boldsymbol{I}$ by scaling $\boldsymbol{Z}$ to $\sqrt{\frac{d}{n}} \boldsymbol{Z}$, the optimization of Eq. 3 is equivalent to achieving feature orthogonalization.

*Proof of Eq. 11.* For any matrix $\boldsymbol{Z} \in \mathbb{R}^{n \times d}$ and $d \leq n$, we can always decompose it using singular value decomposition (SVD) such that $\boldsymbol{Z} = \boldsymbol{U} \boldsymbol{\Sigma} \boldsymbol{V}^\top$, where $\boldsymbol{U}$ is an $n \times n$ orthogonal matrix, $\boldsymbol{\Sigma}$ is an $n \times d$ diagonal matrix with non-negative singular values on its diagonal in non-increasing order, and $\boldsymbol{V}$ is a $d \times d$ orthogonal matrix.

Then, we can rewrite $\log \det \left( \boldsymbol{I} + a \boldsymbol{Z} \boldsymbol{Z}^\top \right)$ as follows:

$$
\begin{aligned}
\log \det \left( \boldsymbol{I} + a \boldsymbol{Z} \boldsymbol{Z}^\top \right) &= \log \det \left( \boldsymbol{I} + a \boldsymbol{U} \boldsymbol{\Sigma} \boldsymbol{V}^\top (\boldsymbol{U} \boldsymbol{\Sigma} \boldsymbol{V}^\top)^\top \right) \\
&= \log \det \left( \boldsymbol{I} + a \boldsymbol{U} \boldsymbol{\Sigma} \boldsymbol{V}^\top \boldsymbol{V} \boldsymbol{\Sigma}^\top \boldsymbol{U}^\top \right) \\
&= \log \det \left( \boldsymbol{I} + a \boldsymbol{U} \boldsymbol{\Sigma}^2 \boldsymbol{U}^\top \right) \\
&= \log \det \left( \boldsymbol{U} (\boldsymbol{I} + a \boldsymbol{\Sigma}^2) \boldsymbol{U}^\top \right) \\
&= \log \det \left( \boldsymbol{U} \right) \det (\boldsymbol{I} + a \boldsymbol{\Sigma}^2) \det \left( \boldsymbol{U}^\top \right) \\
&= \log \det (\boldsymbol{I} + a \boldsymbol{\Sigma}^2),
\end{aligned}
\tag{15}
$$

where the determinant of an orthogonal matrix $\boldsymbol{U}$ is equal to 1, and we used the fact that $\boldsymbol{U} \boldsymbol{U}^\top = \boldsymbol{I}$. Moreover, we can use the properties of the logarithm to prove Eq. 11:

$$
\begin{aligned}
\log \det \left( \boldsymbol{I} + a \boldsymbol{Z} \boldsymbol{Z}^\top \right) &= \log \det \left( \boldsymbol{I} + a \boldsymbol{\Sigma}^2 \right) \\
&= \log \left( \prod_{i=1}^{d} (1 + a \sigma_i^2) \right) \\
&= \sum_{i=1}^{d} \log \left( 1 + a \sigma_i^2 \right).
\end{aligned}
\tag{16}
$$

*Proof of Eq. 12.* The Frobenius norm of a matrix $\boldsymbol{Z}$ can be defined as $\|\boldsymbol{Z}\|_{\mathrm{F}} = \sqrt{\mathrm{Tr}(\boldsymbol{Z}^\top \boldsymbol{Z})}$, where $\mathrm{Tr}(\cdot)$ denotes the trace.

Using the SVD, we can compute the squared Frobenius norm of $\boldsymbol{Z}$:

$$
\begin{aligned}
\|\boldsymbol{Z}\|_{\mathrm{F}}^2 &= \mathrm{Tr}(\boldsymbol{Z}^\top \boldsymbol{Z}) & (17) \\
&= \mathrm{Tr} \left[ (\boldsymbol{U} \boldsymbol{\Sigma} \boldsymbol{V}^\top)^\top (\boldsymbol{U} \boldsymbol{\Sigma} \boldsymbol{V}^\top) \right] & (18) \\
&= \mathrm{Tr}(\boldsymbol{V} \boldsymbol{\Sigma}^\top \boldsymbol{U}^\top \boldsymbol{U} \boldsymbol{\Sigma} \boldsymbol{V}^\top) & (19) \\
&= \mathrm{Tr}(\boldsymbol{V} \boldsymbol{\Sigma}^\top \boldsymbol{\Sigma} \boldsymbol{V}^\top) & (20) \\
&= \mathrm{Tr}(\boldsymbol{V}^\top \boldsymbol{V} \boldsymbol{\Sigma}^\top \boldsymbol{\Sigma}) & (21) \\
&= \mathrm{Tr}(\boldsymbol{\Sigma}^\top \boldsymbol{\Sigma}) & (22) \\
&= \sum_{i=1}^{d} \sigma_i^2, & (23)
\end{aligned}
$$

where we used the properties of trace and the fact that $\boldsymbol{V}$ is orthogonal, so $\boldsymbol{V}^\top \boldsymbol{V} = \boldsymbol{I}$.

## C.3 VISUALIZATION OF FEATURE COVARIANCE MATRIX

In order to gain a visual insight into the role of TCR, we present visualizations depicting the absolute normalized covariance matrix of learned embeddings under different conditions: MIGCL with only the TCR term and the invariance term, MIGCL without the TCR term, and MIGCL applied to the Cora dataset, as illustrated in Figure 4. As can be seen, an evident diagonal structure is discernible within the latent feature space. Specifically, when training exclusively with the TCR term

and the invariance term (as seen in Figure 4(a)), the on-diagonal elements closely approximate 1, while the off-diagonal elements approach 0, indicating a substantial decorrelation of features. In contrast, when the TCR term is omitted, the off-diagonal elements of the covariance matrix exhibit a significant increase, as demonstrated in Figure 4(b). This surge in off-diagonal values suggests that different dimensions fail to capture orthogonal information effectively. Moreover, the MIGCL feature space exhibits a proximity to a diagonal structure, albeit with noisier off-diagonal elements, as depicted in Figure 4(c). This outcome underscores the importance of integrating the structure preservation term and the TCR term within MIGCL. This integration not only enables MIGCL to acquire representations rich in fine-grained graph structure information but also reduces feature redundancy.

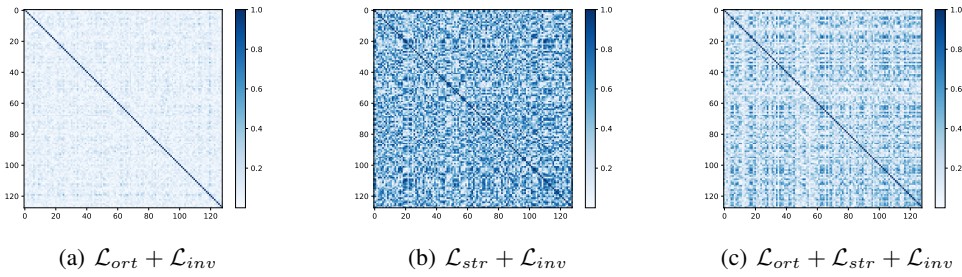

(a) $\mathcal{L}_{ort} + \mathcal{L}_{inv}$   (b) $\mathcal{L}_{str} + \mathcal{L}_{inv}$   (c) $\mathcal{L}_{ort} + \mathcal{L}_{str} + \mathcal{L}_{inv}$

Figure 4: Visualizations of the normalized covariance matrix (absolute value) of learned features on Cora.

## D  ADDITIONAL EXPERIMENTS

### D.1  EFFECT OF THE PROJECTION HEAD

Similar to previous works, such as GRACE (Zhu et al., 2020) and gCooL (Zhu et al., 2020), we also employ a 2-layer MLP projection head. Table 8 demonstrates that MIGCL yields a slight improvement compared to MIGCL without the projection head.

Table 8: Effect of the 2-layer MLP.

| Variants | Cora | | | Photo | | |
|---|---|---|---|---|---|---|
| | MiF1 | NMI | AUC | MiF1 | NMI | AUC |
| MIGCL | 84.47±2.27 | 57.30±0.40 | 94.90±0.43 | 93.21±0.85 | 72.53±0.01 | 97.66±0.09 |
| w/o MLP | 84.10±2.24 | 55.94±0.43 | 94.50±0.57 | 93.08±1.06 | 72.48±0.04 | 97.56±0.09 |

### D.2  EFFECT OF THE DISTORTION MEASURE

Figure 5 depicts the performance of node classification and node clustering across various values of the distortion measure $\epsilon$ in Eq. 2. The results demonstrate the robustness of our model to this hyperparameter. Based on this observation, we set $\epsilon$ to 0.01 for all other experiments.

### D.3  EFFECTS OF AUGMENTATION INTENSITY

We explore investigate the effects of augmentation intensity, specifically $p_{d1}$, $p_{d2}$, $p_{m1}$, and $p_{m2}$, on downstream tasks of node classification and node clustering. Our experiments encompass Cora and Photo datasets, where we vary these parameters from 0.1 to 0.9. To simplify visualization, we set $p_{d1} = p_{d2}$ and $p_{m1} = p_{m2}$. All other hyperparameters remain unchanged as described previously. The results are shown in Figure 6 and Figure 7. The highlighted sections of Figure 6(a) and Figure 7(a) indicate that the performance of node classification remains relatively stable as long as the feature masking ratio is not excessively large. Each dataset exhibits an optimal combination of $(p_d, p_m)$ that enhances generalization for classification and clustering tasks. Overall, our method

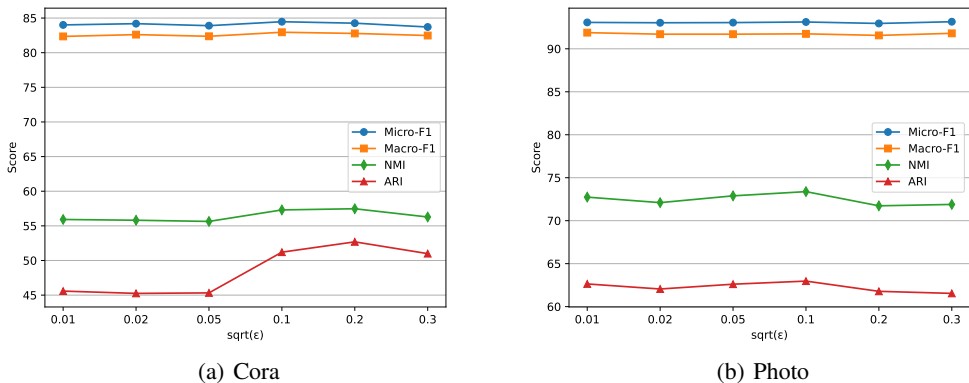

(a) Cora        (b) Photo

Figure 5: Visualization of the effects of $\epsilon$ on Cora and Photo.

demonstrates robustness to augmentation intensity: by maintaining the feature masking ratio and the edge dropping ratio within the appropriate range, our method achieves impressive and competitive performance. Nevertheless, it remains crucial to select a proper augmentation intensity to learn more generalizable and informative representations.

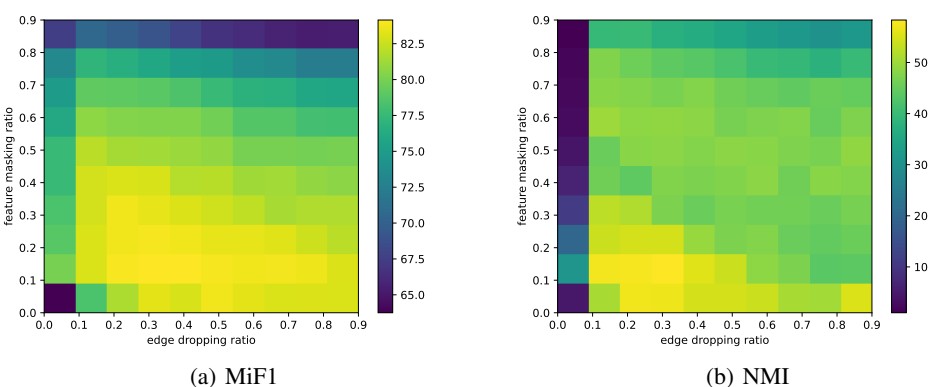

(a) MiF1        (b) NMI

Figure 6: Visualization of the effects of different augmentation intensity on Cora.

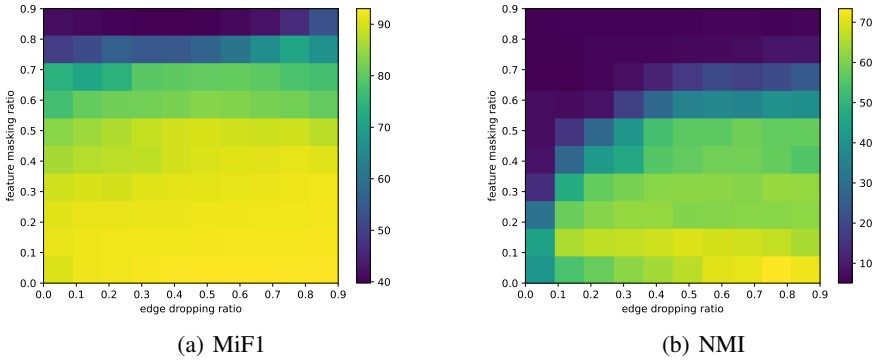

(a) MiF1        (b) NMI

Figure 7: Visualization of the effects of different augmentation intensity on Photo.

### D.4 t-SNE Comparison with Baselines

Due to space limitations in the main text, we provide the t-SNE visualization of all the methods in Figure 8. Here, we reach a similar conclusion to the main paragraph, that MIGCL effectively learns highly structured representations characterized by both inter-cluster separation and intra-cluster compactness.

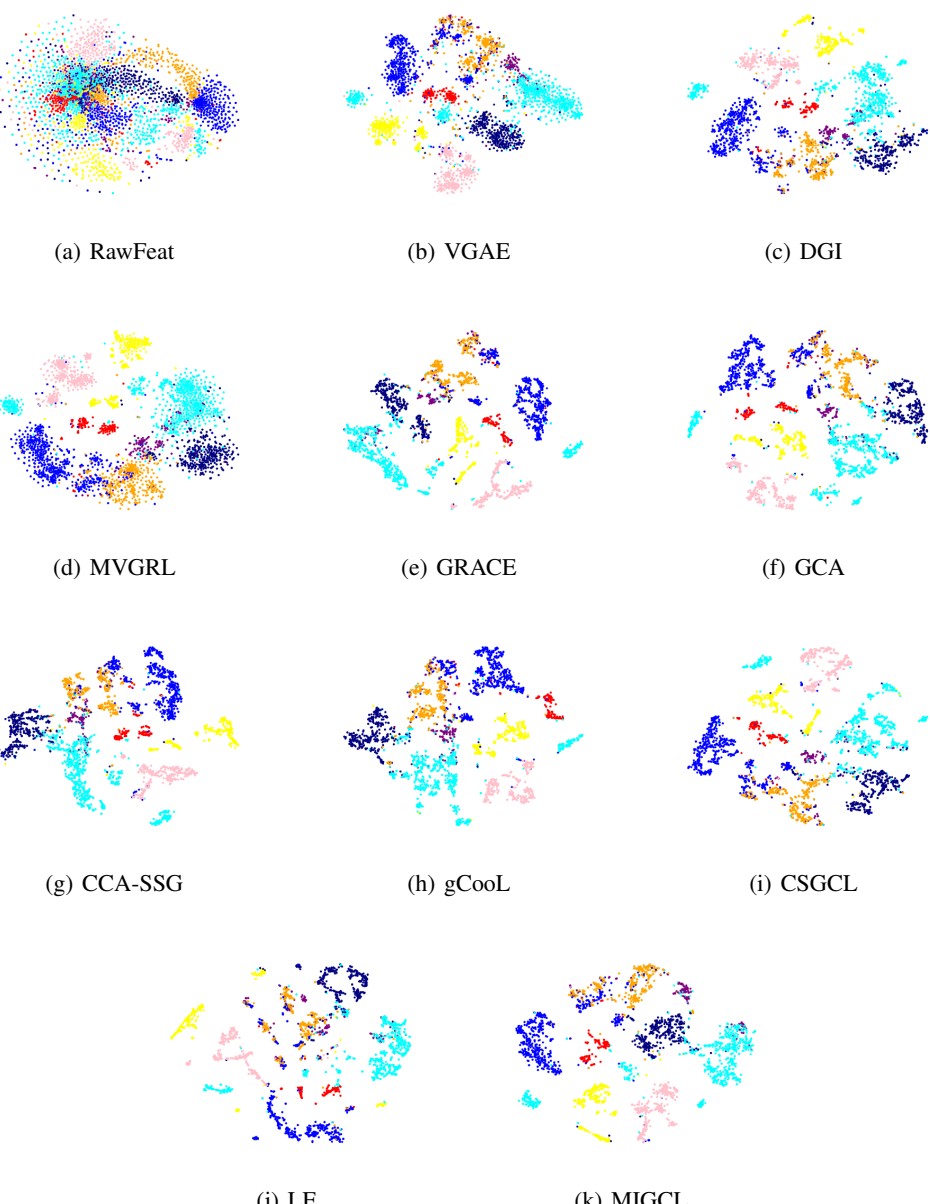

(a) RawFeat      (b) VGAE      (c) DGI

(d) MVGRL      (e) GRACE      (f) GCA

(g) CCA-SSG      (h) gCooL      (i) CSGCL

(j) LE      (k) MIGCL

Figure 8: t-SNE comparison with all baselines on Photo.

## D.5 PERFORMANCE ON OTHER DATASETS

We additionally evaluate the proposed method on two other citation networks: Citeseer and Pubmed (Kipf & Welling, 2017). The experimental results, presented in Table 9, demonstrate that MIGCL still delivers comparable performance, particularly in terms of node classification.

Table 9: Performance of node classification and node clustering on Citeseer and Pubmed (in percentage). OOM indicates ruuning out of memory on a 24GB NVIDIA GeForce RTX 3090 GPU.

| Dataset | Citeseer | | | | Pubmed | | | |
|---|---|---|---|---|---|---|---|---|
| Metric | MiF1 | MaF1 | NMI | ARI | MiF1 | MaF1 | NMI | ARI |
| RawFeat | 58.4±3.0 | 52.2±3.7 | 21.0±3.3 | 17.2±3.4 | 80.7±0.8 | 80.7±0.8 | 31.3±0.2 | 28.1±0.0 |
| LE | 53.0±2.8 | 47.4±2.9 | 23.2±0.1 | 18.5±0.1 | 78.3±0.7 | 76.8±0.9 | 29.4±0.0 | 27.3±0.00 |
| VGAE | 69.2±2.8 | 61.1±2.8 | 39.7±0.0 | 40.9±0.0 | 82.3±0.8 | 81.9±0.9 | 26.8±0.2 | 27.1±0.2 |
| DGI | 70.3±2.1 | 62.7±2.3 | 39.9±0.8 | 40.1±0.8 | 85.7±0.7 | 85.3±0.8 | 24.8±0.0 | 23.9±0.0 |
| MVGRL | 71.9±2.2 | 65.2±2.4 | 43.4±0.1 | 44.1±0.1 | 85.6±0.7 | 85.4±0.8 | **33.8±0.0** | 30.5±0.0 |
| GRACE | 71.3±1.9 | 62.7±1.7 | 40.0±0.3 | 41.3±0.2 | 85.8±0.7 | 85.5±0.7 | 22.0±0.0 | 18.4±0.0 |
| GCA | 71.7±2.3 | 63.5±2.8 | 41.1±0.3 | 41.7±0.3 | 86.3±0.7 | 85.9±0.8 | 31.6±0.1 | 30.8±0.2 |
| CCA-SSG | 72.9±2.3 | 66.0±3.1 | **43.7±0.2** | **44.8±0.3** | 85.4±0.8 | 85.0±0.9 | 31.7±4.7 | 28.3±5.8 |
| gCooL | 72.0±2.1 | 63.0±1.8 | 41.7±0.4 | 42.7±0.5 | 86.5±0.6 | 85.9±0.6 | 33.1±0.0 | 31.9±0.0 |
| CSGCL | 71.6±1.9 | 65.0±2.2 | 42.6±0.7 | 43.0±1.3 | OOM | OOM | OOM | OOM |
| MIGCL | **73.9±3.0** | **67.2±3.0** | 41.2±0.1 | 40.6±0.1 | **86.6±0.6** | **86.0±0.7** | 33.7±0.1 | **32.5±0.1** |

