# OpenReview forum: "Manifold Inspired Graph Contrastive Learning"
_ICLR.cc/2024/Conference — ICLR 2024 Conference Withdrawn Submission_

### Official Review · Reviewer_7ii6 · 2023-10-30

**Soundness:** 3 good
**Presentation:** 2 fair
**Contribution:** 2 fair
**Rating:** 3
**Confidence:** 5

**Summary:**

In this paper, the authors proposed MIGCL, a cutting-edge contrastive representation learning framework that employs cross-view adjacency reconstruction and feature orthogonalization.

**Strengths:**

1. The structure preservation loss is novel for GCL. In this loss function, SGNE is extended to multiview for preserving structural information. In the original GraphSage, SGNE is used to learn unsupervised representation.

**Weaknesses:**

1. I cannot imagine what the relationship between manifold learning and the proposed method. And there are two methods of manifold learning and graph contrastive learning: Contrastive Laplacian Eigenmaps (NeurIPS 2021) and Generalized Laplacian Eigenmaps (NeurIPS 2022). I can understand the relationship between these methods and manifold learning, but in this paper, I cannot follow the story.

2. In this paper, the loss function includes three components. However, two of them are from other methods. This makes the contribution lower than the acceptable bar.

3. The performance is not competitive after combining three loss functions.

**Questions:**

I don't have any questions because I am sure about my conclusion. But I am still open to the discussion with other comments.

---

### Official Review · Reviewer_bWNT · 2023-10-31

**Soundness:** 3 good
**Presentation:** 3 good
**Contribution:** 3 good
**Rating:** 6
**Confidence:** 2

**Summary:**

This paper addresses graph contrastive learning with a novel approach inspired from spectral-based manifold learning techniques.

**Strengths:**

This work is timely, as graph contrastive learning is a hot topic. Moreover, the paper is well written and provides some interesting contributions.

**Weaknesses:**

1 - There are some difficulties in understanding the proposed method. It is not clear how the discrete problem is transformed to a continuous, manifold learning, one. While the resulting optimization problem is inspired from manifold learning, the optimization problem does not have any constraint (besides the conventional orthogonality of Laplacian Eigenmaps). This is weird as the initial problem is a discrete one.

2 - Another issue: While the paper provides some insights on the computational complexity of the proposed method, it would be relevant to provide a comparative analysis within the experiments in order to better position this work within the state of the art.

3 - A related work on self-supervised learning for graphs using manifold learning is the following paper that was not cited by in the paper:
* Sun, Li, Junda Ye, Hao Peng, Feiyang Wang, and S. Yu Philip. "Self-supervised continual graph learning in adaptive Riemannian spaces." In Proceedings of the AAAI Conference on Artificial Intelligence, vol. 37, no. 4, pp. 4633-4642. 2023.
The following paper also investigates a manifold learning perspective for graph contrastive learning:
* You, Yuning, Tianlong Chen, Zhangyang Wang, and Yang Shen. "Bringing your own view: Graph contrastive learning without prefabricated data augmentations." In Proceedings of the Fifteenth ACM International Conference on Web Search and Data Mining, pp. 1300-1309. 2022.
Another paper is
* Sun, Li, Junda Ye, Jiawei Zhang, Yong Yang, Mingsheng Liu, Feiyang Wang, and Philip S. Yu. "Contrastive sequential interaction network learning on co-evolving Riemannian spaces." International Journal of Machine Learning and Cybernetics (2023): 1-17.
It would be relevant to position the contributions of the submitted paper within such related literature

There are many references that are missing journal information, such as Zengyi Li et al., Shengbang Tong et al., …

**Questions:**

Please provide some answers on the points 1, 2, and 3 form Weaknesses, including positioning the contributions of this work within the literature presented in point 3 in Weaknesses.

---

### Official Review · Reviewer_ydcH · 2023-11-02

**Soundness:** 2 fair
**Presentation:** 3 good
**Contribution:** 1 poor
**Rating:** 3
**Confidence:** 4

**Summary:**

The paper introduces MIGCL, a novel approach to graph contrastive learning (GCL) that addresses the challenge of fully leveraging fine-grained graph structure information. MIGCL combines cross-view adjacency reconstruction and feature orthogonalization to capture both node attributes and fine-grained graph structures, preventing informative node representations. Through the use of an information-theoretic objective called Total Coding Rate, MIGCL not only retains fine-grained information but also minimizes feature redundancy, demonstrating its effectiveness and superiority in various graph-related tasks, including node classification, node clustering, and link prediction.

**Strengths:**

By employing the Total Coding Rate and interpreting the model as a practical implementation of the Maximum Entropy Principle, MIGCL provides a strong theoretical foundation, making it a principled and reliable framework for graph representation learning.

**Weaknesses:**

The primary contribution of this paper is the introduction of total rate coding as a regularization term in Graph Contrastive Learning (GCL) to mitigate issues related to dimensional collapse. While the concept of using such regularization in GCL is not entirely new, there are other methods with similar effects, such as Graph Balow Twin [1], SFA [2], and CCA-SSG[3]. However, the author has not thoroughly discussed or compared these approaches.

The proposed method is a straightforward application of the principles outlined in [3], with no significant technical challenges. The author should provide more in-depth insight and analysis regarding the extra advantages of their proposed method over existing alternatives. It is essential to clarify why one should opt for the proposed method rather than the established method.

Furthermore, as the author's work draws motivation from manifold learning, it is pertinent to include references to related work that discusses manifold learning and graph representation learning, such as [5] and [6]. This would help provide a more comprehensive context for the reader.


[1] Bielak, Piotr etal. Graph Barlow Twins: A self-supervised representation learning framework for graphs KBS 2022

[2] Zhang, Yifei, et al. "Spectral feature augmentation for graph contrastive learning and beyond AAAI 2023

[3] Zhang, Hengrui, et al. "From canonical correlation analysis to self-supervised graph neural networks. NeurIPS 2021

[4] Li, Z., Chen, Y., LeCun, Y., & Sommer, F. T. Neural manifold clustering and embedding Arxiv 2020

[5] Zhu, Hao, and Piotr Koniusz. "Generalized laplacian eigenmaps."  NeurIPS 2022

[6] Zhu, Hao, and Piotr Koniusz."Contrastive Laplacian Eigenmaps"  NeurIPS 2021

**Questions:**

see weakness

---

### Official Review · Reviewer_iU8b · 2023-11-06

**Soundness:** 2 fair
**Presentation:** 2 fair
**Contribution:** 2 fair
**Rating:** 3
**Confidence:** 4

**Summary:**

This paper introduces a graph contrastive learning approach, incorporating a contrastive objective function inspired by manifold learning principles. In particular, the contrastive objective function comprises three crucial components: 1. The "cross-view adjacency reconstruction term" which preserves fine-grained graph structure information. 2.The "feature orthogonalization term" designed to eliminate redundant correlations among features, preventing collapsed representations. 3. The "view alignment term" that ensures that the learned representations remain invariant to augmentations of the same node instance.

**Strengths:**

1.	The paper investigates the issue of feature collapse in graph contrastive learning, which is a fundamental and significant problem in this field.
2.	The paper conducts a comprehensive set of main experiments, specifically evaluating the model's performance on three sub-tasks: Node Classification, Node Clustering, and Link Prediction.

**Weaknesses:**

1.	The motivation of the paper is unclear. The authors argue that a good contrastive learning objective function should involve three essential ingredients: an invariance criterion, collapse avoidance, and structure preservation. In the methodology, the authors propose a feature orthogonalization term to achieve collapse avoidance and a cross-view adjacency reconstruction term to achieve structure preservation. However, I believe that the authors have not adequately claimed two critical points: 1). Regarding the collapse avoidance issue, the authors do not summarize existing methods to solve this problem and why these approaches may not yield satisfactory results. 2). For structure preservation, the authors mention that "These studies achieve structure preservation either implicitly or at a rudimentary level" and consider "fine-grained structure" as important. The authors have not theoretically or experimentally demonstrated why "fine-grained structure" is crucial for GCL. Additionally, how is "explicit" and "implicit" defined, and what is "fine-grained"? Why is the performance of existing works considered unsatisfactory? Additionally, there is an important related work [1] that has not been mentioned, even though it is related to recommendations. The GCL methods it introduces bear a notable resemblance to the concept of structure preservation presented in this paper.
2.	The method lacks strong innovation as it appears to be a combination of multiple existing works (lacking novelty) and lacks a solid theoretical foundation. For instance, the Total Coding Rating (TCR) seems to be a direct reproduction of existing work without any innovative adaptations for the current scenarios. My concern is that relying solely on TCR seems to result in an expansion of the feature space. How can you ensure the compactness within the same class [2]? Furthermore, what advantages does it offer compared to existing relevance methods in the field of (CL) [3]?
3.	Eq. 5 does not explain how global information is retained and how the balance between local and global information is achieved. This is a crucial aspect of the paper's innovation. This formula seems to serve a similar purpose as the method mentioned in [1], with the only difference being that the author has implemented it in a cross-view setting.
4.	The experiments do not provide a detailed explanation of the contributions of this paper, such as the extent of feature collapse in existing related works and how the paper's method is demonstrated to capture fine-grained graph structural information.

[1] Lin Z, Tian C, Hou Y, et al. Improving graph collaborative filtering with neighborhood-enriched contrastive learning[C]. Proceedings of the ACM Web Conference 2022. 2022: 2320-2329.
[2] Chan K H R, Yu Y, You C, et al. Deep networks from the principle of rate reduction[J]. arXiv preprint arXiv:2010.14765, 2020.
[3] Zbontar J, Jing L, Misra I, et al. Barlow twins: Self-supervised learning via redundancy reduction[C]//International Conference on Machine Learning. PMLR, 2021: 12310-12320.

**Questions:**

Please refer to the comments above.

---

### Official Review · Reviewer_kqKA · 2023-11-10

**Soundness:** 3 good
**Presentation:** 3 good
**Contribution:** 2 fair
**Rating:** 5
**Confidence:** 4

**Summary:**

The paper presents a paradigm for learning vectorial representations from graph-based data with the aim of incorporating both feature attributes of nodes, as well as node connections. They try to benefit from the self-supervision offered by off-the-shelf graph augmentation techniques and choose a contrastive approach for learning representations.


The authors break down the task into designing a loss function that has 3 elements to reflect 3 different aspects of a successful graph representation learning and then try to elaborate on and justify each part accordingly. The loss elements, as the authors categorize them, are as follows:

Firstly, in order to promote view-invariance, they maximize cosine similarity of two augmented views of the same node-feature pair. This is done similarly to many other works.

Secondly, in order to avoid trivial solutions and collapsed representations, they propose to maximize a surrogate for the entropy of representations, which is chosen as Total Coding Rate (TCR), as was also chosen in the work of [Liu et al. 2022a] under the context of contrastive learning in vision. This is supposed to promote orthogonality, and hence the authors draw analogies of their work to the classical work of Laplacian Eigenmaps (LE). The authors consider this part as a contribution.

Thirdly, in order to maximally encode"fine-grained structure information", the authors propose to use "cross-view adjacency" reconstruction, which enforces the similarity of embeddings of nodes that are directly connected, and penalizes similarities of non-connected nodes. The paper considers this as an important contribution of the work.


The authors then make an effort to provide a "theoretical insight", by associating their proposal to the maximum entropy principle.



In conjunction with some graph augmentation, the proposed loss function is then used to train vectorial representations for the nodes of a given graph by using standard deep learning recipes. These node embeddings can then be used in some down-stream tasks.

The paper presents experiments on 6 public graph databases on the tasks of node classification, node clustering and link prediction, and compares the proposed method with 10 contenders. They initially show superior performance on all measures on 4 datasets and later in the appendix (D.5), they show performance which is comparable to other existing methods. They then perform ablation studies to examine the effect of each of the components of the proposed loss function, justifying their choice.

**Strengths:**

- The paper is rather well-structured, is very easy to follow, and the descriptions are rather clear. There seems to be rather sufficient references to relevant prior works.

- It addresses an important problem; that of benefitting from contrastive learning within graph representation learning, as was proven to be largely successful in vision and NLP. Moreover, it is somehow along the recent trend of incorporating insights from classical methods of node embedding (like the LE method) into the modern GNN's literature, which sounds like a promising trend and valid research question.

- The break-down of the loss into 3 components each with an intuitive motivation behind is quite nice.

- The ablation studies are very useful and quite convincing of the choice of their loss components within their experiments.

**Weaknesses:**

- The novelty of the method is rather limited. While it is very valid to incorporate ideas from the vision communities into graph representation learning, the mre adoption of MEC of[Liu et al. 2022a] from vision contrastive learning to graph contrastive learning is not highly novel. Unless supported by extensive experiments and on scale, this level of contribution is quite average for ICLR.

- The experiments are carried out only on 6 datasets from the DGL family, and they are rather small to medium-sized ones. It would be useful for the readers to know how the proposed method applies to larger-scale data (e.g., like Reddit, ZINC, ..), or some more examples of the medium-sized family. If the method doesn't perform favorably on those, the paper should mention this, and perhaps try to justify. As a more general comment, the graph representation learning community is highly active with dozens of works proposed at every venue each claiming to be the SOTA (e.g., take a look at the current submissions for ICLR'24). By not choosing a very coherent set of datasets with identical evaluation setups (e.g., as would be guaranteed by using the standardized benchmarks like the OGB), it is very difficult for the community to draw clear conclusions on the merits of each work.

- The paper labels itself as "manifold-inspired'', yet the direct connection to the manifold learning literature is missing. They only very briefly mention in the introduction about "graphs being discrete analogs of continuous manifolds", but what particularly makes this paper a manifold-learning method is unclear. It seems like the connection to the manifold-learning techniques is rather on heuristics, which applies  to a wide range of methods proposed in this domain.

- Similar to the above, the paper draws some analogies to the classical work of Laplacian Eigenmaps (LE). However, the direct connection with this method is not clear. LE is a spectral method based on the Laplacian of the graph with a closed form solution consisting of the d eigenvectors corresponding to the d smallest eigenvalues. The proposed solution does not use graph Laplacian and uses graph adjacency, and its solution is found iteratively using standard gradient-based methods within deep learning. This makes very little resemblance between the two. It seems like the only resemblance of the two methods is in pushing for orthogonal representations, where in the proposed method it is enforced by the TCR, while in the LE, it is guaranteed by construction.

- The paper claims to benefit from "fine-grained structure information'' using cross-view adjacency reconstruction, while other methods are described as"not explicitly focusing on capturing'' such information, or achieving structure preservation "either implicitly or at a rudimentary level". It is not clear where this claim comes from. The loss term $\mathcal{L}_{str}$ is essentially taking only first-order connections into account as it promotes similarity only if two nodes are directly connected by an edge, and is penalizing similarity of nodes that are not  directly connected. This first-order connection is present in many works.

**Questions:**

- In the work of MEC [Liu et al. 2022a], the TCR(Z1, Z2) loss is a function of both Z1 and Z2, which essentially push for both view-invariance, as well as orthogonality in the same expression, while the current work maximizes TCR(Z1) and TCR(Z2), independently from each other, and enforces view-invariance separately. What is the reason for this choice? Can't the loss function be simplified by merging the two?

- Again in comparison with MEC, why is the expensive log-det term used, rather than its equivalent trace-based expression, or the Taylor series-based approximation proposed in [Liu et al. 2022a]? Perhaps some low order Taylor expansions will coincide with other prior works. It would be interesting to check this.

- The idea of incorporating Total Coding Rate and more generally its interpretation from the point of view of the maximum entropy principle can be categorized as "information theoretically inspired heuristics". What is the relation of this proposition to other information theoretic heuristics within the same context like the Infomax principle? The reader would, e.g., want to know what are the differences between maximizing entropy of description and maximizing mutual information between some components, as is done in some other works.

- The loss-term $\mathcal{L}_{str}$ promotes similarity of nodes only if they are directly connected, and penalizes similarity of embeddings as soon as they are not directly connected. While the authors also acknowledge that it is not a very intuitive choice (and hence justify it later by ablation studies), it would be interesting to see what would happen if this aggressive penalization was somehow loosened. For example, in the second term of eq. (5) in the paper, the authors could choose a more relaxed version than $j \notin \mathcal{N}_i$.